# Fault-Tolerant Multilevel Converter to Feed a Switched Reluctance Machine

**Vítor Fernão Pires [1,2,*]** , **Armando Cordeiro [1,2,3]** , **Daniel Foito [1,4]** and **Armando J. Pires [1,4]**

1 SustainRD, EST Setubal, Polytechnic Institute of Setúbal, 2910-761 Setúbal, Portugal; acordeiro@deea.isel.ipl.pt (A.C.); daniel.foito@estsetubal.ips.pt (D.F.); armando.pires@estsetubal.ips.pt (A.J.P.)
2 INESC-ID, Polytechnic Institute of Lisboa, 1700-001 Lisboa, Portugal
3 ISEL, Polytechnic Institute of Lisboa, 1700-001 Lisboa, Portugal
4 CTS-UNINOVA, Polytechnic Institute of Setúbal, 2829-516 Caparica, Portugal
* Correspondence: vitor.pires@estsetubal.ips.pt

**Abstract:** The switched reluctance machine (SRM) is one of the most interesting machines, being adopted for many applications. However, this machine requires a power electronic converter that usually is the most fragile element of the system. Thus, in order to ensure high reliability for this system, it is fundamental to design a power electronic converter with fault-tolerant capability. In this context, a new solution is proposed to give this capability to the system. This converter was designed with the purpose to ensure fault-tolerant capability to two types of switch faults, namely open- and short-circuit. Moreover, apart from this feature, the proposed topology is characterized by a multilevel operation that allows improvement of the performance of the SRM, taking into consideration a wide speed range. Although the proposed solution is presented for an 8/6 SRM, it can be used for other configurations. The operation of the proposed topology will be described for the two modes, fault-tolerant and normal operation. Another aspect that is addressed in this paper is the proposal of fault detection and diagnosis method for this fault-tolerant inverter. It was specifically developed for a multilevel SRM drive. The theoretical assumptions will be verified through two different types of tests, firstly by simulation and secondly by experiments with a laboratory prototype.

**Keywords:** switched reluctance machine (SRM); fault-tolerant operation; fault detection; fault diagnosis; multilevel topology

## 1. Introduction

Electric motors are fundamental in today's industrialized world. There are several electrical machine types [1–3]. However, one that is considered very interesting for many applications is the switched reluctance machine. The adoption of this machine is due to the fact that it is characterized by a very simple structure, robustness, provides a large starting torque and a large speed range, is rare-earth-material free, and presents a high efficiency. In this way, it has been adopted for many applications, such as in electric and hybrid vehicles [4–6], aircraft and aerospace systems [7–10], home appliances [11], industrial tools [12], wind generators [11–16], pumping systems [17–19], mining [20] and storage systems [21,22]. However, a very important aspect is that this machine requires a power electronic converter. So, apart from the motor, the design and choice of the power electronic converter are also fundamental for the drive.

For SRM drives, several power electronic converters have been proposed [23]. Many of the applications are based on two-level converters [24–28]. Through the use of these, topologies two voltage levels will be generated, allowing for the magnetization or demagnetization of the machine windings. However, other solutions have been used and proposed, such as the multilevel topologies and other solutions based on the impedance source [29–34]. Several aspects should be considered for the choice of the power converter topology, such as the speed range, torque ripple, and performance, among others. Apart

from this, another aspect that is very important in several applications is the reliability of the system.

The power electronic converter of an SRM system is usually the weakest element regarding reliability. The most typical faults in these converters are the power transistors' open- and short-circuit faults. These faults can severely affect the operation of the machine. In order to overcome these problems, some fault-tolerant schemes based on hardware or software have been used. These fault-tolerant schemes usually rely on redundant components, such as switches and relays [35,36]. Further, fault detection and tolerant systems have been considered very important in a huge number of applications [37–40]. The same is applied to the systems that are included in the SRM machine and correspondent drive.

Several two-level converters with fault-tolerant capabilities have been proposed to be used with the SRM. Most of the solutions are based on the use of redundant switches in which they will replace the power semiconductor in the open- or short-circuit faults, as can be seen in [41–43]. Meanwhile, with the purpose to reduce the number of backup power semiconductors, a solution was proposed in [44] with the addition of two controlled switches and six thyristors. However, all these solutions only allow providing fault tolerance to the open-switch fault. Thus, other solutions that also provide fault tolerance against open- and short-circuit faults were also presented. These solutions require extra power semiconductors and relays [45–49]. Another approach to this issue was through the use of multilevel power electronic converters. Apart from their interesting features regarding their use in applications where SRMs with high-speed range is needed, they also provide some fault tolerance, which is fundamental in safety-critical applications. A neutral point clamped (*NPC*) multilevel topology that was used to provide multiple voltage levels showed that it also ensures some fault-tolerant capability [50–52]. However, it presents some limitations regarding some faults, namely when the open-circuit faults are in the inner power semiconductors. Thus, some other solutions were proposed, namely through the use of full *NPC* and T-Type topologies [53,54]. However, they still have some limitations, namely regarding multiple faults and the maximum voltage magnetization and demagnetization in the leg with the fault.

Another important aspect related to the fault-tolerant converters is the requirement of fault-detection algorithms. This is also another aspect that has been explored. Several methods have been proposed to achieve this requirement. Some of the approaches are based on the frequency domain. One of the approaches is developed through the use of spectrum analysis [55,56]. Another approach for the detection of a switch fault was developed through the wavelet packet decomposition with current reconstruction [57]. However, several other approaches were not based on the frequency domain. An approach in which the detection of a switch fault phase is performed through the determination of the winding resistance of the SRM was also proposed. This approach uses an extended Kalman filter for the determination of that resistance [58]. A more simplified method in which the average values of the motor winding currents were used was also presented in [59]. A method that also uses the winding motor currents but is based on the current Park's vector was also presented [60,61]. Another approach, based on the winding currents, was proposed in [62], in which the comparison of the freewheeling current of different phase currents is used. A method that is based on the entropy feature approach was presented in [63]. Another method, based on the motor windings currents, was presented in [64], but this method also requires the measurement of the DC-bus currents. Methods that are based on the converter voltages were also proposed. A method based on the analysis of the motor winding voltage patterns was presented in [65]. A method based on the motor winding voltages signature was also presented in [66]. To mention that the described methods were only applied to SRM two-level voltage converters. In reality, practically all the research works have been performed for these kinds of converters by which fault-detection methods for multilevel drives for SRM are almost inexistent.

In the context of the need for reliable power electronic converters for the SRM, a new topology with fault-tolerant capability is proposed in this paper. The proposed fault-

tolerant converter can handle power semiconductors in open- and short-circuit faults without losing performance capability. On the other hand, it is also able to handle some multiple faults. Apart from this, this scheme is also indicated to SRMs with high-speed range since it also provides multilevel operation. The clarification of the proposed scheme will be verified through an analysis of an 8/6 SRM under different types of faults. Additionally, it will be proposed a fault diagnostic method for the proposed fault-tolerant topology. The particularity of this fault-detection method is that it is applied to a SRM multilevel power electronic converter. Moreover, the theoretical analysis will be confirmed through several simulations and laboratory tests.

## 2. Proposed Multilevel Fault-Tolerant Converter

Many multilevel fault-tolerant topologies have been developed over the last years, presenting different characteristics and capabilities. One of the power converter topologies that presents some fault-tolerant capability and is highly indicated to be used in applications that require a high-speed range is the *NPC* Asymmetric Half-Bridge (*NPC AHB*). The topology of this power converter applied to an 8/6 SRM is presented in Figure 1, where it is possible to see that an extra middle voltage level can be applied to the motor windings through the connection of those windings to the capacitors middle point. Multilevel topologies are also suitable for medium-voltage, high-power applications. For such applications, the number of voltage levels of the multilevel converter can increase in order to satisfy the requirements regarding the voltage level. It is also possible to increase the power of the electric drive using multiple parallel branches or structures to achieve higher currents. In this way, this topology can be used for high-power SRM drives, maintaining the same principles presented in this paper.

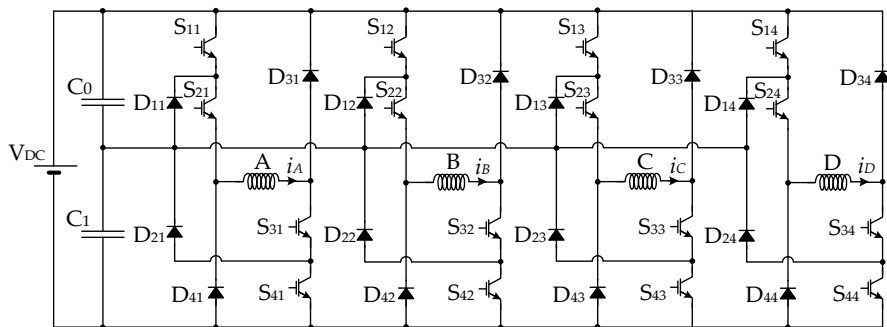

**Figure 1.** Neutral Point Clamped Asymmetric Half-Bridge (*NPC-AHB*) applied to a 8/6 SRM.

Studying the multilevel fault-tolerant topology shown in Figure 1, it is evident that if there is a short-circuit in one of the outer switches (e.g., $S_{11}$, $S_{41}$, $S_{12}$, $S_{42}$, ... ), then the converter is still able to operate. Nevertheless, the middle voltage level ($V_{DC}/2$) is not possible to be applied anymore in the faulty phase. On the other hand, in the case of an open-circuit fault, the maximum voltage that can be applied is the middle voltage level ($V_{DC}/2$). So, this could affect the magnetization process and the performance of the SRM drive. In case of a short-circuit fault in one of the inner switches (e.g., $S_{21}$, $S_{31}$, $S_{22}$, $S_{32}$, ... ), then it will happen to the opposite of the outer switches, losing the full voltage ($V_{DC}$). Regarding this topology, the major problem is the open-circuit fault in one of the inner switches. In this case, it is not possible anymore to magnetize the motor winding. So, for this kind of fault, the circuit does not provide fault-tolerant operation.

In order to provide full fault-tolerant capabilities to the power converter topology of the SRM drive, a new topology, based on the *NPC-AHB* presented in Figure 1, is proposed. In Figure 2, it is possible to see the proposed power converter topology connected to an 8/6 SRM. It is possible to see that the original *NPC-AHB* topology was changed by adding active switches to the inverter's clamping diodes ($S_{5j}$ and $S_{6j}$), $j \in \{1, 2, 3, 4\}$. These new devices combined with another group of power switches and diodes connected to each

branch of the converter ($S_{7j}$ and $S_{8j}$) allow new current paths to recover lost voltage levels due to short- or open-circuit faults in the main power switches. To achieve the desired operation, it also introduced a bidirectional solid-state relay (*SNP*) to isolate the neutral point (*NP*) during the fault-tolerant operation. This device is essential to the operation of the proposed solution. Finally, two *NC* (Normally Closed) mechanical relays ($K_{11}$, $K_{12}$, $K_{21}$, $K_{22}$, ... ) in each branch are used to isolate any short-circuit fault, allowing recovery of lost voltage levels without creating additional short-circuit in the capacitors.

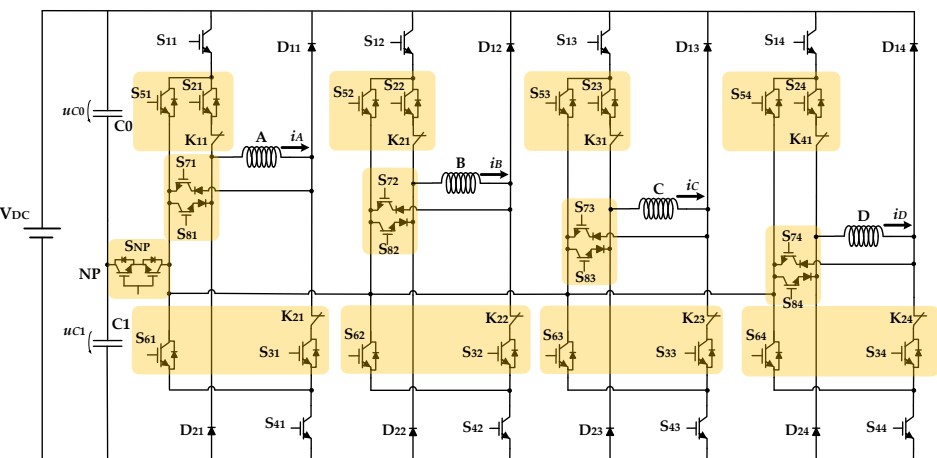

**Figure 2.** Fault-tolerant solution based on the *NPC-AHB* topology applied to an 8/6 SRM.

Several examples of failure modes and how to proceed with the proposed topology are presented next. Figure 3 shows, as an example, an open-circuit fault in the power device $S_{21}$. As stated earlier for the original *NPC-AHB*, this failure mode in the inner devices usually leads to loss of full ($V_{DC}$) and middle voltage ($V_{DC}/2$), and the SRM drive will only operate with $n$-1 windings with consequent torque reduction and high ripple. Using the proposed solution is possible to mitigate such faults and recover the full and middle voltage. To obtain the full voltage (see Figure 3a), it is necessary to first isolate the connection to the *NP* using the solid-state relay *SNP*. This allows creating a path to achieve again the full voltage in phase *A* of the SRM through the power devices $S_{11}$, $S_{51}$ and $S_{81}$. To obtain the middle voltage, it is only necessary to disconnect the power devices $S_{11}$ and $S_{51}$ and connect again the solid-state relay *SNP* (see Figure 3b). The demagnetization process is similar to the normal operation (see Figure 3c).

Figure 4 illustrates, as another example, an open-circuit fault in the power device $S_{11}$. Considering the same strategy mentioned to the previous fault, it is also possible to recover the full voltage of phase *A*. In this case, there are different paths that can be used to recover such voltage, but they require the use of power devices from other branches (or legs). This is not critical regarding overloads since the selected devices are disconnected when phase *A* is operating. Since the operation of the SRM requires that during the demagnetization of one phase and magnetization of the next phase (in the sequence) should be both in operation, the choice must be performed through the next available phase, not in use. This means that the use of the adjacent phases is possible, but this is not advisable since, in this condition, the inner power switches must be designed to withstand a reverse voltage higher than the full voltage, which is not desirable. In this example, phase *C* is not adjacent to phase *A* and must be used to do this operation (see Figure 4). In this case, there are two possible paths to recover the full voltage after disconnecting the *SNP* device: first, -$S_{13}$; $S_{53}$; $S_{81}$; second, -$S_{13}$; $S_{53}$; $S_{21}$. The strategy to obtain the middle level ($V_{DC}/2$) and the demagnetization process are the same as presented in Figure 3b,c, respectively.

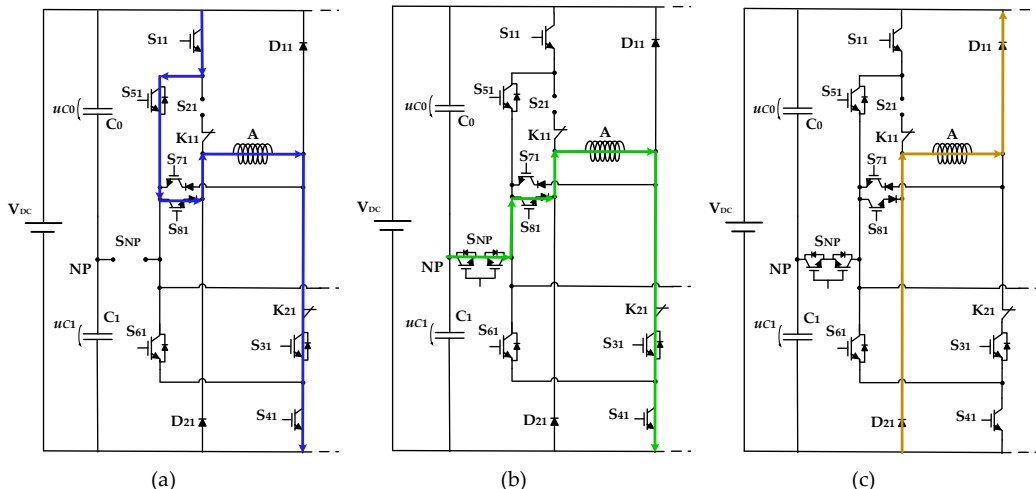

**Figure 3.** Fault-tolerant solution based on the *NPC-AHB* topology applied to an 8/6 SRM. Example of an open-circuit fault in power device $S_{21}$ and recover path to obtain the full (**a**), middle (**b**) voltage of phase *A* and (**c**) demagnetization process.

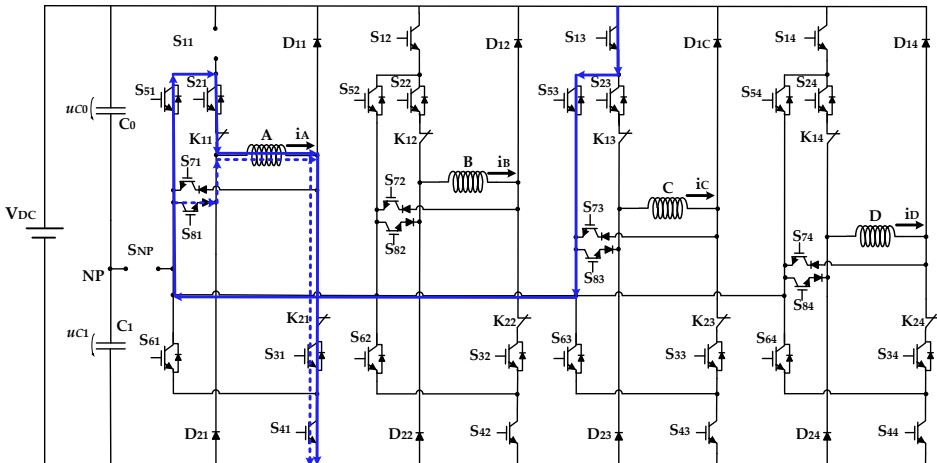

**Figure 4.** Fault-tolerant solution based on the *NPC-AHB* topology applied to an 8/6 SRM. Example of an open-circuit fault in power device $S_{11}$ and recover paths to obtain the full voltage of phase *A*. Two possible paths are available to achieve the full voltage.

The proposed solution is also able to deal with multiple failures in the same branch. In the example presented in Figure 5, a simultaneous open-circuit fault in $S_{11}$ and $S_{21}$ can be seen. Similar to the previous situation presented in Figure 4, it is also possible to recover the full voltage of phase *A* using the power devices of phase *C* after disconnecting the *SNP* devices: $S_{13}$; $S_{53}$; $S_{81}$. In this case, there is a single path that can be used to recover such voltage.

A short-circuit failure mode is now illustrated in Figure 6. In this example, it is a short-circuit fault in the power device $S_{11}$. In this situation, it is not possible to impose the middle level ($V_{DC}/2$) since whenever $S_{21}$ is connected to the full level ($V_{DC}$), it is applied due to the short-circuit fault of the power device $S_{11}$. In order to minimize the impact of this short-circuit fault, it is necessary to isolate the faulty device and faulty branch. The proposed solution to isolate short-circuit faulty devices is based on the use of mechanical relays. In this example, the relay $K_{11}$ should be used. Although a typically slow response (5–10 ms), the mechanical relays are an economical solution and are expected to be operated once after the fault occurs (normally closed contacts switch over to an open state in the proposed topology). Obviously, during the interval it takes to open the relays, it will not

be possible to have complete control over the load. It is worth noting that the relays used in the proposed scheme operate in a "soft switch" mode, i.e., not extinguishing currents but instead promoting their deviation to alternative paths. Additionally, the mechanical relays do not consume energy in the normal operation of the drive (without device failure, the relays are disconnected). After the isolation of the faulty branch, the operation is similar to the one presented in Figure 5, and a single path is available after disconnecting the *SNP* devices: $S_{13}$; $S_{53}$; $S_{81}$. The strategy to obtain the middle level ($V_{DC}/2$) and the demagnetization process is the same as presented in Figure 3b,c, respectively.

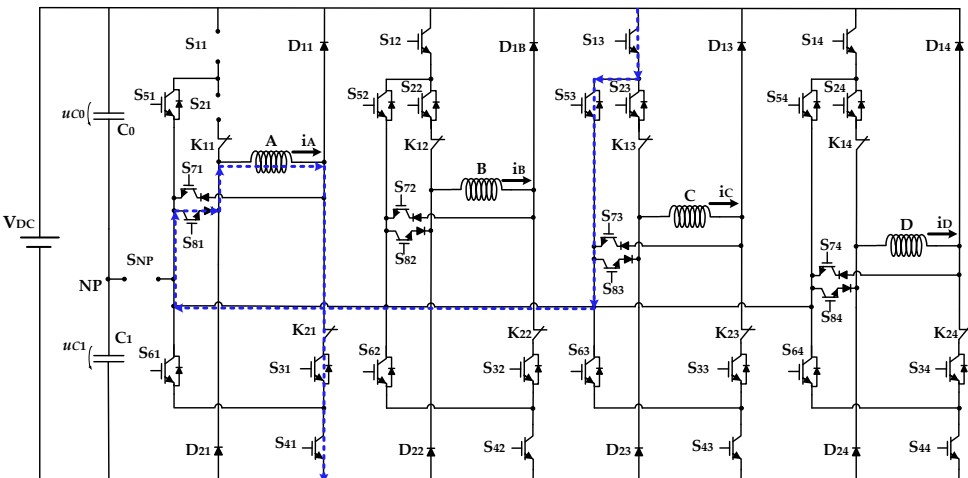

**Figure 5.** Fault-tolerant solution based on the *NPC-AHB* topology applied to an 8/6 SRM. Example of a multiple open-circuit fault in power device $S_{11}$ and $S_{21}$. A single possible path is available to achieve the full voltage.

In the case of a short-circuit in power device $S_{21}$, the behavior of the converter would be the continuous imposition of the middle level ($V_{DC}/2$) in phase *A*. This will lead to difficulties in the demagnetization process since the diode $D_{21}$ cannot turn on to reverse the voltage in winding *A*. The solution to the fault in this device is the same as the short-circuit in power device $S_{11}$; namely, the isolation of the branch and using exactly the same procedures.

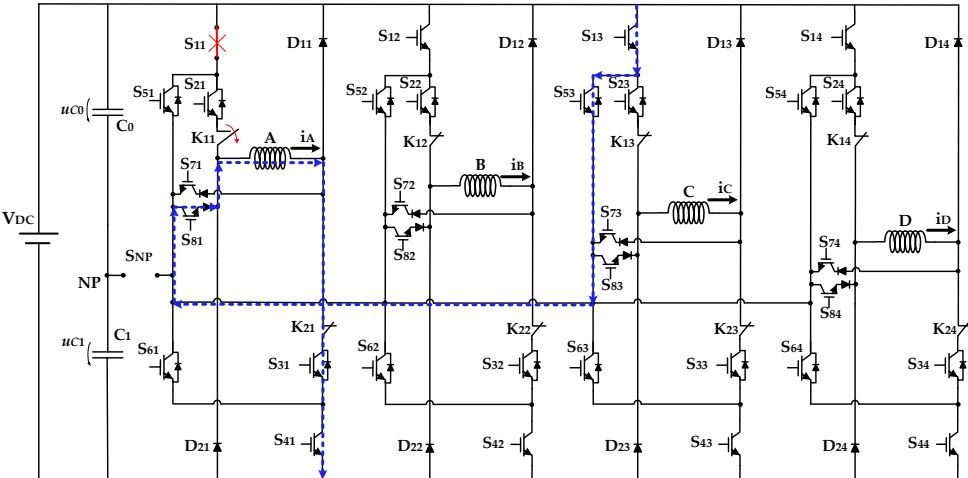

**Figure 6.** Fault-tolerant solution based on the *NPC-AHB* topology applied to an 8/6 SRM. Example of a short-circuit fault in power device $S_{11}$ and recover path to obtain the full voltage of phase *A*. A single path is available after isolation using the mechanical relay $K_{11}$.

Despite not being described and exemplified in this section, the same approach can be adopted for the faults in power semiconductors of other phases. The proposed solution is also fault-tolerant to multiple faults in power devices of different phases. As an example, it is possible to have an open-circuit fault in the power devices $S_{11}$ and $S_{12}$ (or $S_{12}$ and $S_{13}$) at the same time. Nevertheless, in this situation, the inner switches must be designed to support higher reverse voltages. Because the phases of the SRM operate independently (and only two at a time during magnetization and demagnetization) and with a predefined sequence, it is possible to explore different paths (see Figure 5) to recover lost voltage levels. Although this scheme has been presented for the 8/6 SRM, it can be extended for other machines with a different number of phases.

### 3. Control of the Drive and Balance of DC Voltage Capacitors

There are several schemes that can be used to control the speed of the SRM machine. In this work, the control system presented in the diagram of Figure 7 was adopted. The current controller and correspondent modulator are the base of most of the control systems used for SRM machines. In this case, the modulator includes a fault-detection block and a fault logic decision block integrated with a voltage balance block. More details of the control strategy can be seen in Figure 8 and explained in detail next.

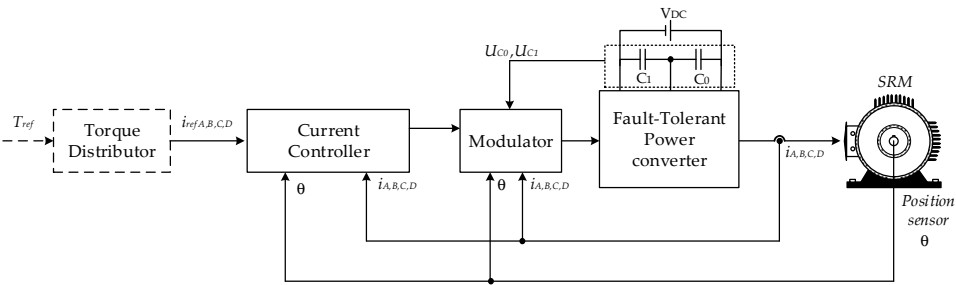

**Figure 7.** Simplified block diagram of the SRM speed controller.

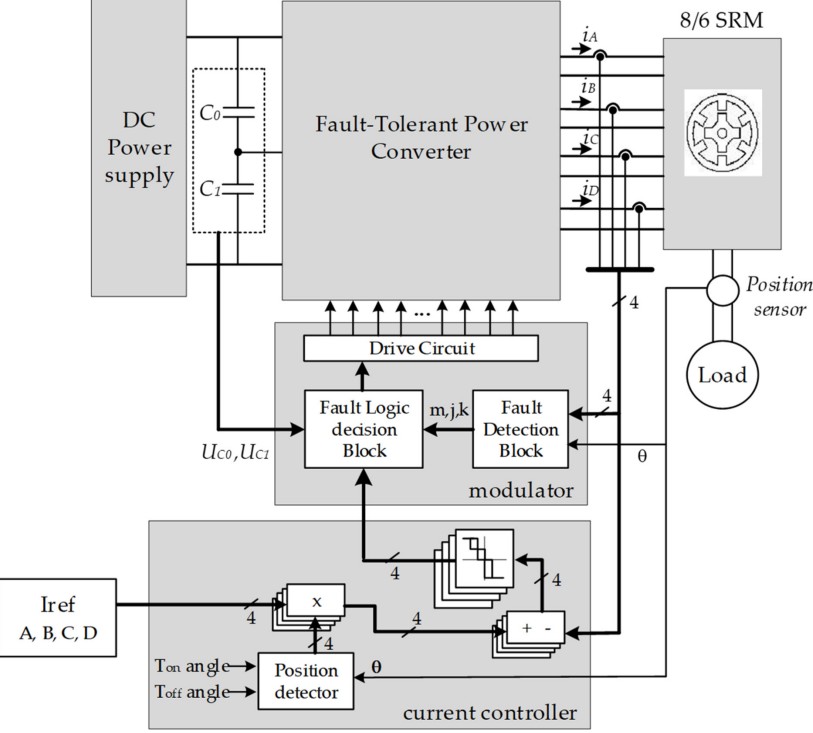

**Figure 8.** Detailed block diagram of the SRM speed controller (where the fault-tolerant power converter is the topology presented in Figure 2).

In Figure 8, θ represents the rotor position, $m \in \{0, 1, 2, 3, 4\}$ indicates de switch number inside each phase, $j \in \{1, 2, 3, 4\}$ indicate the phase number (or phase *A*, *B*, *C* and *D* in the 8/6 SRM); *k* indicates if the failure mode is an open-circuit (*k* = 0) or short-circuit (*k* = 1).

The current controller, which regulates the phase current, will be implemented by a current hysteresis controller. Since the fault-tolerant power converter is based on the original *NPC-AHB*, the use of a multilevel hysteretic comparator is proposed. Thus, in order to generate the three possible voltage levels, a three-level comparator will be used, as presented in Figure 9. So, the applied voltage level for each phase will be a function of the current error (Equation (1)), and the applied voltage will increase with the increase in the current error. According to the current error, a switching variable for each phase will be defined as $\gamma_j \in \{+1, 0, -1\}$.

$$\begin{cases} if\ i_{ref\ j} - i_j > +2\Delta i \Rightarrow V_{phase\ j} = +V_{DC} \Rightarrow \gamma_j = +1 \\ if\ i_{ref\ j} - i_j > +\Delta i \Rightarrow V_{phase\ j} = +\frac{V_{DC}}{2} \Rightarrow \gamma_j = 0 \\ if\ i_{ref\ j} - i_j > -\Delta i \Rightarrow V_{phase\ j} = -V_{DC} \Rightarrow \gamma_j = -1 \end{cases} \quad (1)$$

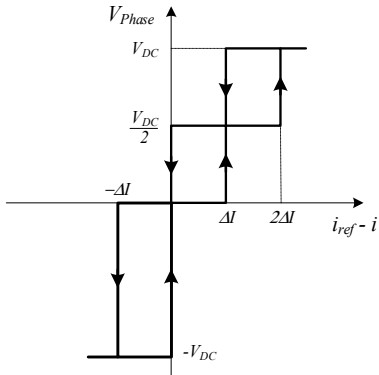

**Figure 9.** Proposed multilevel comparator for the current controller.

Another aspect related to the control system is the generation of the gate signals that are a function of the current error, failure mode of power devices, and voltage of *DC* capacitors. This is performed inside of a modulator block connected to the output of the hysteretic comparator. In normal operation, there is only one possible combination for each full voltage level ($-V_{DC}$ and $+V_{DC}$) and for the inner voltage levels ($+V_{DC}/2$) there are two possible combinations using the upper and lower capacitors (see Table 1). In the case of the inner voltages, the adopted switching combination must be chosen to ensure the balance of the applied voltage to the *DC* capacitors. There are two ways to ensure this balance, or through the right combination of the switches that control the phase under operation or through the switches that control one of the phases that are not in operation. This is possible due to the phase independence of this type of motor. However, in this case, the balance will be made through the switches that control the phase under operation. In order to balance the voltage across the capacitors, a control law associated with the capacitor voltages will be considered. This law is given by (Equation (2)), which consists of the difference between the measured capacitors voltages.

$$e_{V_{DC}} = V_{DC\_capacitor\_upper} - V_{DC\_capacitor\_lower} \quad (2)$$

**Table 1.** Generic voltage levels in normal operation for the *j* phase.

| State (ST) | $S_{1j}$ | $S_{2j}$ | $S_{3j}$ | $S_{4j}$ | SNP | $e_{V_{DC}}$ | m,j | k | Voltage Level (VL) |
|---|---|---|---|---|---|---|---|---|---|
| 1*j* | 1 | 1 | 1 | 1 | 1 | x | 0,*j* | x | $+V_{DC}$ |
| 2*j* | 1 | 1 | 1 | 0 | 1 | <0 | 0,*j* | x | $+V_{DC}/2$ |
| 3*j* | 0 | 1 | 1 | 1 | 1 | >0 | 0,*j* | x | $+V_{DC}/2$ |
| 4*j* | 0 | 0 | 0 | 0 | 1 | x | 0,*j* | x | $-V_{DC}$ |

x—do not care.

Table 1 presents the generic voltage levels for each *j* phase in normal operation, considering the necessary switches (only main power switches) and respective states to balance the *DC* capacitors. In this table *m* = 0 indicates the absence of failures, and thus the value of *k* is not taken into consideration. From this table is possible to see that only the states 2*j* and 3*j* are able to balance the capacitors. This is performed in every cycle and healthy phase.

Considering the proposed fault-tolerant control strategy and desired operation of the SRM drive, it is possible to create a generic table for each *j* phase with all the available states of the switching devices regarding the normal operation and the faulty modes, namely the open-circuit (*OC*) and the short-circuit (*SC*). This information can be seen in Table 2.

**Table 2.** Summary of possible switching states in normal and faulty operation for each generic *j* phase.

| ST | $S_{1j}$ | $S_{2j}$ | $S_{3j}$ | $S_{4j}$ | $S_{5j}$ | $S_{6j}$ | $S_{7j}$ | $S_{8j}$ | $S_{1s}$ | $S_{4s}$ | $S_{5s}$ | $S_{6s}$ | SNP | VL |
|---|---|---|---|---|---|---|---|---|---|---|---|---|---|---|
| 1*j* | 1 | 1 | 1 | 1 | 0 | 0 | 0 | 0 | 0 | 0 | 0 | 0 | 1 | $+V_{DC}$ |
| 2*j* | 1 | 1 | 1 | 0 | 0 | 0 | 0 | 0 | 0 | 0 | 0 | 0 | 1 | $+V_{DC}/2$ |
| 3*j* | 0 | 1 | 1 | 1 | 0 | 0 | 0 | 0 | 0 | 0 | 0 | 0 | 1 | $+V_{DC}/2$ |
| 4*j* | 0 | 0 | 0 | 0 | 0 | 0 | 0 | 0 | 0 | 0 | 0 | 0 | 1 | $-V_{DC}$ |
| 5*j* | 0 | 1 | 1 | 1 | 0 | 0 | 0 | 0 | 1 | 0 | 1 | 0 | 0 | $+V_{DC}$ |
| 6*j* | 0 | 0 | 1 | 1 | 0 | 0 | 0 | 1 | 1 | 0 | 1 | 0 | 0 | $+V_{DC}$ |
| 7*j* | 1 | 0 | 1 | 1 | 1 | 0 | 0 | 1 | 0 | 0 | 0 | 0 | 0 | $+V_{DC}$ |
| 8*j* | 0 | 0 | 1 | 1 | 0 | 0 | 0 | 1 | 0 | 0 | 0 | 0 | 1 | $+V_{DC}/2$ |
| 9*j* | 0 | 1 | 1 | 1 | 0 | 0 | 0 | 0 | 0 | 0 | 0 | 0 | 1 | $+V_{DC}/2$ |
| 10*j* | 1 | 1 | 0 | 1 | 0 | 1 | 1 | 0 | 0 | 0 | 0 | 0 | 0 | $+V_{DC}$ |
| 11*j* | 1 | 1 | 0 | 0 | 0 | 0 | 1 | 0 | 0 | 1 | 0 | 1 | 0 | $+V_{DC}$ |
| 12*j* | 1 | 1 | 1 | 0 | 0 | 0 | 0 | 0 | 0 | 1 | 0 | 1 | 0 | $+V_{DC}$ |
| 13*j* | 1 | 1 | 0 | 0 | 0 | 0 | 1 | 0 | 0 | 0 | 0 | 0 | 1 | $+V_{DC}/2$ |
| 14*j* | 1 | 1 | 1 | 0 | 0 | 0 | 0 | 0 | 0 | 0 | 0 | 0 | 1 | $+V_{DC}/2$ |

The *s* variable used in the identification of the switches in Table 2 represents *s* = *j* + 2 and obey congruence modulus four arithmetic within their respective domains, i.e., after the last value comes the first one again. Taking as an example, if the fault is in phase *j* = 1 then the switches that must be used to recover the lost voltage levels must be the devices of phase 3 or phase *C* (*s* = *j* + 2 = 3). This means that if a fault occurs in phase *j* = 4 then the switches that must be used are from phase 2 or phase *B*. This happens because the next phase after *D* is again phase *A*, then phase *B*, and so on.

Based on the provided information, is now possible to create a decision table after fault detection, according to the faulty device *m*, phase *j*, switching variable $\gamma_j$ and type of failure (*OC* or *SC*). The possible states must be chosen according to such variables and are presented in Table 3. The switching devices according to the defined states can be found in Table 2. Note that it is possible to find some redundant states in the case of open-circuit failure modes. It is also possible to see that balancing the *DC* capacitors during the operation of the faulty phase is not possible. This is not a problem since it is possible to perform the balance using the healthy phases, as presented in Table 1. In other words, during the operation of the faulty phase (even in fault-tolerant mode) it is not possible to balance the *DC* capacitors since the bidirectional *SNP* device is disconnected. Nevertheless,

in the operation of the next healthy phases, the bidirectional *SNP* device is connected again, and the *DC* voltage balance capability returns. Notice that the damaged phase is ruled by Table 3 and the healthy phases by Table 1 (where voltage balance is available), and the control strategy must switch between these tables. This can be explained using an example. After the detection of an open-circuit failure in the power device *S*21 (this device is identified by $m = 2$ a $j = 1$ as the second device of phase 1 or phase *A*), the control strategy chose in Table 3 the state $m,j = (2,1)$ for each switching variable $\gamma_j$, which are the states *6j* or *7j* for $\gamma_j = +1$, *8j* for $\gamma_j = 0$ and *4j* for $\gamma_j = -1$. Since $j = 1$ then the effective states are 61, 71, 81 and 41, which have correspondence to the power devices of Table 2. In this situation, the state 81 is used to impose $+V_{DC}/2$ through the capacitor $C_1$. Since this is the unique state possibility, it is not possible to assure the voltage balance of *DC* capacitor. However, since the next phase in operation is phase *B* and considering the absence of failure, the control strategy must choose the state *2j* or *3j* (22 or 32 since $j = 2$ in this case) of Table 1 according to $e_{V_{DC}}$, allowing again the voltage balance capability. The same principle can be applied to all the healthy phases.

**Table 3.** Fault decision table.

| *m,j* | $\gamma j$ | $e_{V_{DC}}$ | State (ST) $K = 0$ (*OC*) | State (ST) $K = 1$ (*SC*) |
|---|---|---|---|---|
| 1,*j* | +1 | x | 5*j*; 6*j* | 6*j* |
| 1,*j* | 0 | x | 8*j*; 9*j* | 8*j* |
| 1,*j* | −1 | x | 4*j* | 4*j* |
| 2,*j* | +1 | x | 6*j*; 7*j* | 6*j* |
| 2,*j* | 0 | x | 8*j* | 8*j* |
| 2,*j* | −1 | x | 4*j* | 4*j* |
| 3,*j* | +1 | x | 10*j*; 11*j* | 11*j* |
| 3,*j* | 0 | x | 13*j* | 13*j* |
| 3,*j* | −1 | x | 4*j* | 4*j* |
| 4,*j* | +1 | x | 11*j*; 12*j* | 11*j* |
| 4,*j* | 0 | x | 13*j*; 14*j* | 13*j* |
| 4,*j* | −1 | x | 4*j* | 4*j* |

In case of short-circuit faults, it is also necessary to switch the mechanical relays to isolate the faulty half branch. This operation is based on a simple logic equation given by (Equation (3)). This equation indicates that the corresponding relay should operation whenever a short-circuit fault occurs in each half branch.

$$\begin{cases} K_{1j} = \left(S_{1j\_SC}\right) \ or \ \left(S_{2j\_SC}\right) \\ K_{2j} = \left(S_{3j\_SC}\right) \ or \ \left(S_{4j\_SC}\right) \end{cases} \tag{3}$$

The proposed control strategy is also able to handle multiple faults inside each power converter phase. To do this operation, another decision table was created, establishing priorities according to the faulty severity. This is an important aspect since the fault-detection block only returns a single faulty device and failure mode for each phase. By creating this strategy, it is possible to filter the most critical fault to provide the best fault-tolerant result. This priority is given by Table 4. For example, in case of a simultaneous open-circuit failure in the switches $S_{14}$ and $S_{24}$, the fault-detection block will generate the value $(m,j,k) = (1,4,0)$, indicating that the most critical failure is in the switch $S_{14}$. The priority list was based on the analytical performance of the converter in the fault-tolerant mode. In the case of an open-circuit of an outer device (e.g., $S_{11}$), the $+V_{DC}$ voltage will not be possible to apply, and the desired current phase will not be possible to establish. Notice that the $+V_{DC}/2$ voltage available will not be enough to raise the current to the desired value. This will generate high degradation of the SRM torque. This was considered as priority one. In the case of an open circuit of any inner device (e.g., $S_{12}$), the $+V_{DC}/2$ voltage will not be possible to apply, but it is always possible to use the $+V_{DC}$ voltage.

In this situation, it is possible to raise the current to the desired value, but because the intermediate voltage is not available, the switching frequency tends to increase during this period. Nevertheless, the operation of the drive is lightly affected, with reduced impact on the SRM torque. This was considered priority two, which is less important than the previous fault. As a result of any short-circuit inner device (e.g., $S_{12}$), the converter will tend to impose a fixed $+V_{DC}/2$ voltage in the faulty phase. However, due to the presence of the bidirectional *SNP* device and the mechanical relay, it is possible to isolate the faulty device (and faulty branch) and operate the converter similarly to an inner open-circuit failure. This was considered as priority three since the impact on the SRM torque is quite similar to the proposed priority two. The main reason to select this failure as priority three is the fact that after isolation, it becomes similar to an inner open-circuit failure, which has a higher priority. The last critical failure mode was considered the short-circuit in an outer device (e.g., $S_{11}$). In this situation, the converter will tend to impose a fixed $+V_{DC}$ voltage in the faulty phase. Since the inner power device is connected in series with the faulty device, it is always possible to disconnect the inner device and minimize this effect. One problem of this situation is the impossibility to apply the $+V_{DC}/2$ voltage without the creation of a short-circuit in the capacitors. Again, due to the use of a mechanical relay, it is possible to isolate the faulty device and operate the converter in a fault-tolerant mode similar to an open-circuit failure in the inner device. Due to these reasons, this last situation was considered as priority four.

**Table 4.** Priority list for multiple failure modes inside each phase.

| Priority | Multiple Failure Modes |
|----------|------------------------|
| 1 | Open circuit in the outer devices |
| 2 | Open circuit in the inner devices |
| 3 | Short-circuit in the inner devices |
| 4 | Short-circuit in the outer devices |

## 4. Fault-Detection Scheme for the Proposed Fault-Tolerant Drives

As described in the introduction, several fault-detection schemes have been developed for the SRM drives. For this drive, a fault-detection scheme based on the work [54] was developed. A detection scheme based on the current's patterns, at which symmetry indexes will be associated, is proposed.

In order to identify an open-switch fault of the inverter, the creation of indexes is proposed, which will be related to the motor winding currents pattern. These indexes will be related to the symmetry or asymmetry of the patterns of these currents. Due to this, these indexes will be designated as symmetry index (*SI*). For the computation of these indexes, an approach that is based on the entropy feature is used. For this computation, $n$ samples taken over one cycle (sliding window) are also considered. Thus, from the acquired data set, the corresponding entropy at instant $p$ is first determined in accordance with:

$$H_k(i) = -\sum_{j=1}^{n} \frac{1}{n}|i_k(j)| \log_2\left(\frac{1}{n}|i_k(j)|\right), \ k = A, B, C, D \qquad (4)$$

However, with the purpose to obtain fault indexes that are independent of the operation of the motor, load and other factors, four normalized severity indexes (considering that in this case, the number of motor windings is four) will be defined in accordance with (7) that are functions of Equations (4)–(6).

$$IAV_k = 4H_k(i), \ k = A, B, C, D \qquad (5)$$

$$IAV_T = \sum_{k=A}^{D} H_k(i) \qquad (6)$$

$$SI_k = \frac{IAV_k}{IAV_T} \tag{7}$$

These four normalized indexes can now be used to detect and identify a switch fault. In this way, for the condition of all switches being healthy, all the normalized indexes will give a value of one. However, if there is a fault in one of the switches, those normalized indexes will change their values. So, for one of the upper switches with an open fault, the index associated with that leg will change, becoming lower than one (but higher than zero). The other indexes also change, but for a higher value than one. In the case of the lower switch, the index associated with the leg of that switch will change to zero. Regarding the other ones, they will change for a value higher than one. In the case of a switch with a short-circuit fault, the behaviour of the normalized indexes will be inverse. Thus, for this fault type, the index associated with the leg with the switch under fault will increase its value. The other normalized indexes (associated with the legs with healthy switches), will change to values that will be below one.

## 5. Simulation Results

The proposed fault-tolerant multilevel converter for the SRM was tested through computer simulations. For these simulations, the well-established Matlab/Simulink program with the Simscape Power Systems library was used. The multilevel power converter under tests was designed to operate a four-phase 8/6 SRM and was fed by one *DC* source with 300 V connected to two capacitors of 100 μF. On the other hand, the tests were performed with the motor operating at a speed of 1500 rpm.

With the goal to verify the fault-tolerant capability of the proposed power electronic converter, several tests with different fault types were performed. For the first test, an open-switch fault was considered, namely for the semiconductor $S_{12}$, (affecting directly the SRM phase *B*). The results of this test are presented in Figure 10, being possible to verify that until 0.695 s, the converter operates in normal mode, but after that instance, the open fault appeared. As expected, this fault will affect the maximum positive voltage level, by which the current in phase *B* of the SRM is affected. Due to that, the current in that phase does not reach the reference value. Through these results, it is also possible to confirm that after 0.715 s, there was a reconfiguration of the circuit by which the malfunction of the converter originated by the open-switch fault disappeared. Another test with an open-switch fault, but for a different power semiconductor and a different phase, was also performed. In this case, the fault was for the inner switch $S_{21}$ and for phase *A*, being possible to verify the obtained results in Figure 11. As can be seen, after 0.695 s the open-switch fault appeared, by which the voltage and current of the winding associated with the leg of the faulty switch were affected. In this case, since the fault was in the inner switch, no voltage at all is applied to that motor winding. This is the same for the winding current that remains zero all the time. However, at $t = 0.742$ s, the fault-tolerant mode was applied in which first the connection to the *NP* was isolated using the solid-state relay *SNP*, and after that, a path through the power devices $S_{81}$ was created. The obtained results confirm that after the reconfiguration, the circuit recovers its initial operation, namely applying the required voltage and current to the affected motor winding. Another aspect that was tested was related to the voltage balance between the *DC* capacitors. In Figure 12, those voltage capacitors for both open-switch tests ($S_{12}$ and $S_{21}$) are presented. As it is possible to verify by these figures, during the fault, there is some unbalance between those voltages (more visible for the $S_{12}$ fault), but in fault-tolerant mode is recovered (same average value).

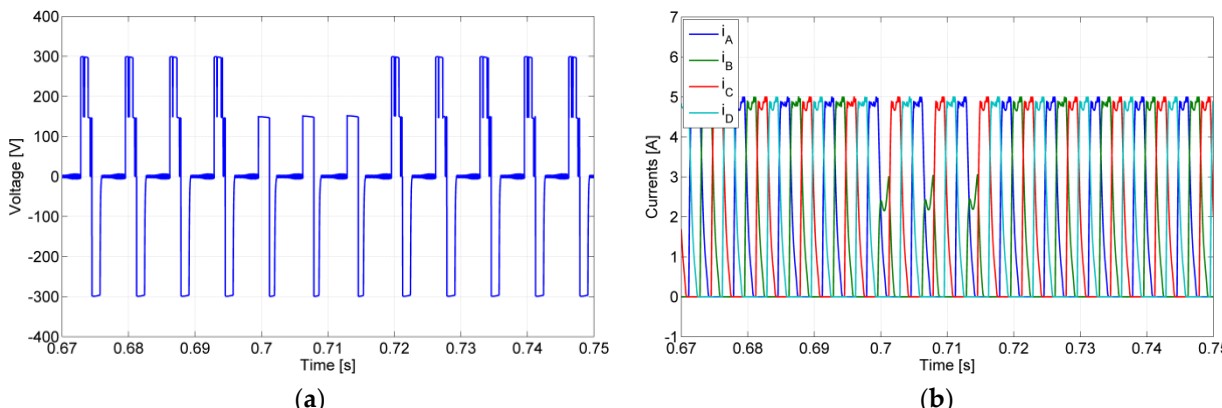

**Figure 10.** Simulation test with the converter operating in normal mode until $t = 0.695$ s and after that with an open-switch fault in the semiconductor $S_{12}$ (**a**) winding voltage of phase $B$; (**b**) winding currents. Fault-tolerant mode after $t = 0.715$ s.

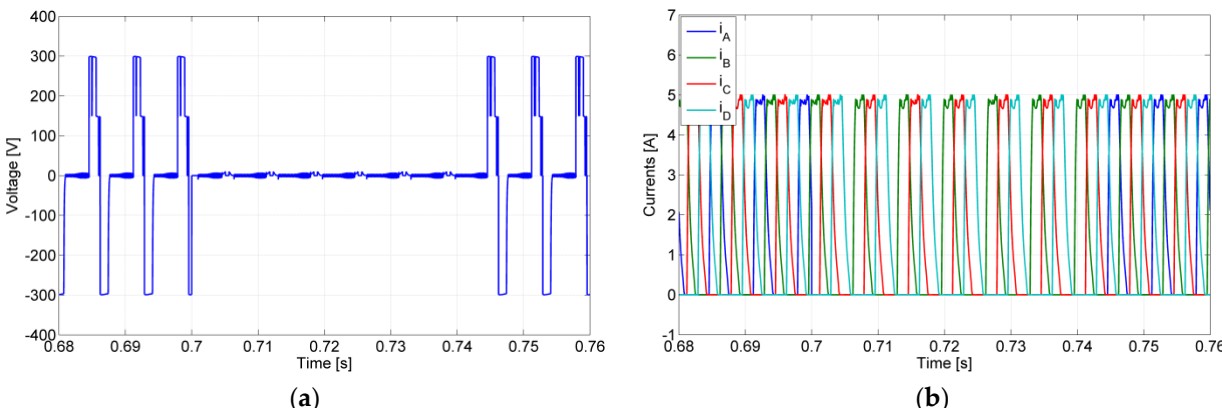

**Figure 11.** Simulation test with the converter operating in normal mode until $t = 0.695$ s and after that with an open-switch fault in the semiconductor $S_{21}$ (**a**) winding voltage of phase $A$; (**b**) winding currents. Fault-tolerant mode after $t = 0.742$ s.

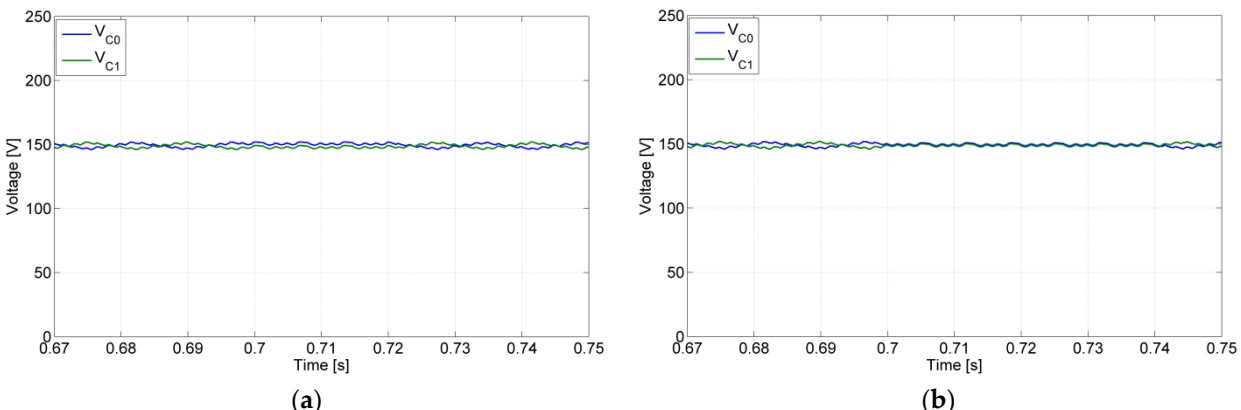

**Figure 12.** Simulation test with the converter operating in normal mode until $t = 0.695$ s and after that with an open-switch fault (**a**) voltage capacitors for open-switch fault in the semiconductor $S_{12}$; (**b**) voltage capacitors for open-switch fault in the semiconductor $S_{21}$. Fault-tolerant mode after $t = 0.742$ s.

Another aspect that was verified was the fault-detection and diagnosis algorithm. In Figure 13, the obtained results for the four normalized indexes are presented. These results show the capability of this method. In fact, through Figure 13a, it is possible to see that

during normal and fault-tolerant operation, all of these indexes present a value of one. However, during the open-switch fault of the semiconductor $S_{12}$, the index associated to the leg with the faulty switch changes to approximately 0.6, while the other ones change to approximately 1.12. This confirms what is expected from these indexes for this fault. In Figure 13b, the same indexes but for an open-switch fault in the semiconductor $S_{21}$ are presented. Again, outside of the fault period, all the indexes are equal to one. However, during the fault, the index associated with the leg with the switch fault changes to zero, while the other ones change to approximately 1.33. Again, this result confirms what was expected from this detection algorithm.

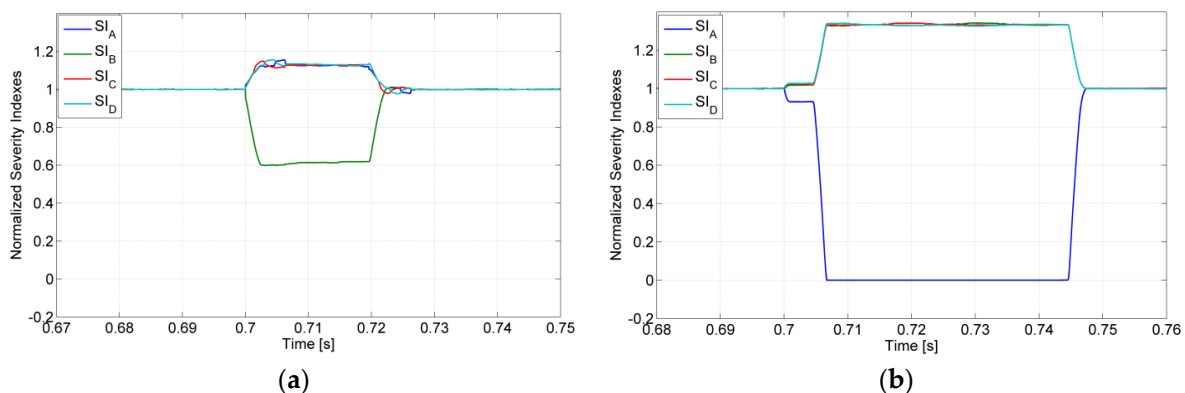

(**a**)                                                                (**b**)

**Figure 13.** Results of the four normalized indexes (**a**) for an open-switch fault in the semiconductor $S_{12}$; (**b**) for an open-switch fault in the semiconductor $S_{21}$.

Tests for a short-circuit switch fault were also performed. In Figure 14, the result for this kind of test is presented, in which the switch under fault was the $S_{24}$. Through the analysis of this figure, it is possible to verify that this fault only affects the operation during the demagnetization period (fault in $t = 0.7$ s). In fact, due to this fault, it is not possible to apply the maximum voltage to the winding associated with the leg with the switch under fault, by which the current decreases more slowly. However, after the reconfiguration of the circuit (fault-tolerant mode), at $t = 0.73$ s, this limitation is removed. After this instant, the maximum negative voltage is again applied to the motor winding associated with the leg under fault. In this way, the current will decrease again in a fast way, ensuring a fast demagnetization. The balance of the *DC* voltage capacitors is lost only during the fault, although not in a critical way. As can be seen in Figure 15, outside of the failure period, this balance is achieved, and the fault-tolerant mode is operating as desired.

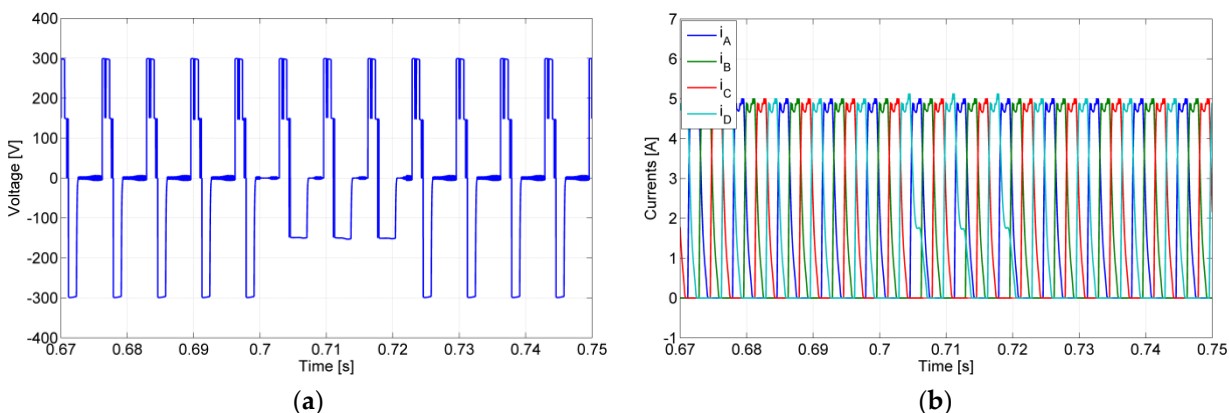

(**a**)                                                                (**b**)

**Figure 14.** Simulation test with the converter operating in normal mode until $t = 0.7$ s and after that with a short-circuit fault in the semiconductor $S_{24}$ (**a**) winding voltage of phase *D*; (**b**) winding currents. Fault-tolerant mode after $t = 0.73$ s.

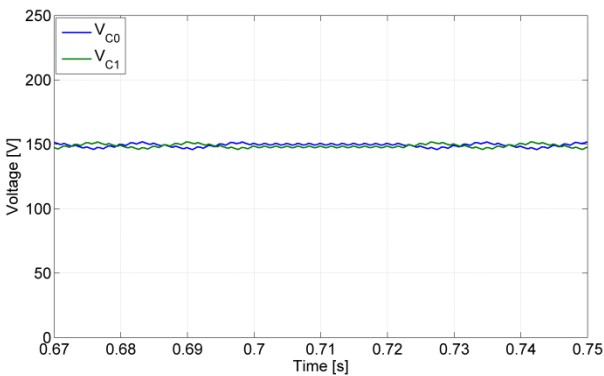

**Figure 15.** Voltage capacitors obtained from the simulation test with the converter operating in normal mode until $t = 0.7$ s and after that with a short-circuit fault in the semiconductor $S_{24}$. Fault-tolerant mode after $t = 0.73$ s.

The behavior of the fault-detection and diagnosis algorithm for the short-circuit fault in the switches was also verified. In Figure 16, the obtained results for the four normalized indexes are presented. Again, it is possible to see that during normal and fault-tolerant operation, all these indexes present a value of one. However, during the short-circuit fault of the semiconductor $S_{24}$, the index associated with the leg with the faulty switch changes to approximately 1.2, while the other ones change to approximately 0.93. This confirms what is expected, namely that in this fault type, the indexes change in an inverse way from what happens under the presence of an open-circuit failure mode.

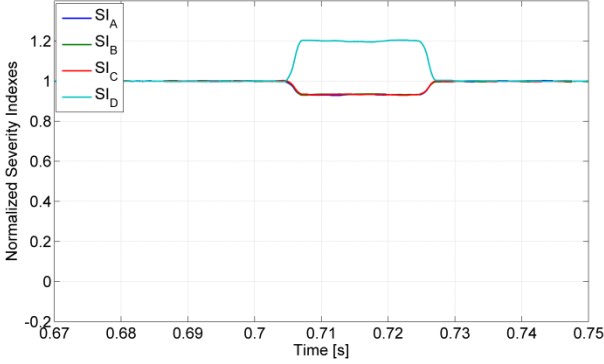

**Figure 16.** Results of the four normalized indexes for a short-circuit fault in the semiconductor $S_{24}$.

Tests associated with multiple faults were also developed. From Figure 17, it is possible to verify the obtained results by multiple faults. So, in $t = 0.7$ s, there was an open-switch fault in semiconductor $S_{11}$. As expected, the magnetization process was affected since it is no longer possible to apply the maximum voltage. At $t = 0.72$ s, there is a second fault, namely an open-switch fault in semiconductor $S_{21}$. As a consequence, after this instant, it is no longer possible to apply any voltage to the motor winding associated with the leg under fault. However, through the reconfiguration of the circuit, it is possible to recover completely the normal operation of the drive. This reconfiguration is performed through the operation of the solid-state relay $SNP$ and leg associated with the motor winding $C$. So, the middle voltage will be ensured through $SNP$ and semiconductor $S_{81}$. The full voltage is ensured through the operation of semiconductors $S_{13}$ and $S_{53}$ (leg of the motor winding $C$). This reconfiguration (fault-tolerant mode) is performed at $t = 0.74$ s. In fact, it is possible to verify (see Figure 17) that after the instant, the voltage and current associated with the motor winding are established, where the converter recovers completely.

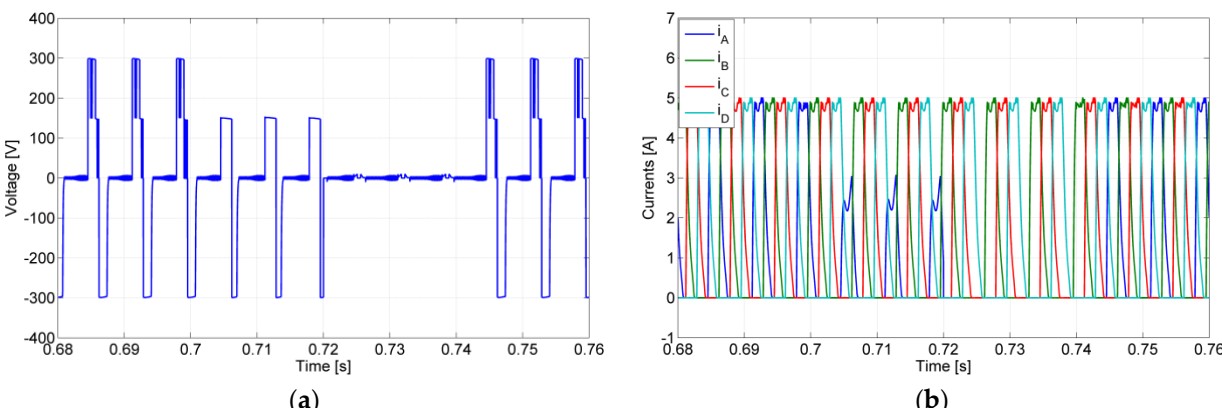

(**a**)  (**b**)

**Figure 17.** Simulation test with the converter operating in normal mode until $t$ = 0.7 s, an open-switch fault ($S_{11}$) after that instant and another open-switch fault ($S_{21}$) at $t$ = 0.72 s (**a**) winding voltage of phase *A*; (**b**) winding currents. Fault-tolerant mode after $t$ = 0.74 s.

## 6. Experimental Results

The verification of the simulation tests was also performed by experimental tests. For these tests, a laboratory prototype was used with the same parameters as the ones used in the simulation. The control of the SRM drive was developed through a *DSPACE* tool. This tool was used with the purpose to implement several decision tables and to deal with all the operation modes (normal, failure and fault-tolerant modes) and also voltage capacitors balance. Several instruments and sensors were used in this research. Regarding the instruments, an oscilloscope TDS3014C with 100 mV/A current probes and voltage probes with 1 V/100 V. Regarding the sensors, four 20 A (*LEM HSX20-NP*) current sensors were used to measure the current of each phase and to send the value for the DSPACE1104, which controls the speed of the motor. The position signal of the SRM rotor is measured with an absolute encoder (TWK DAB 66-M 360 W C01 with 0–10 V per revolution), which was also sent to the DSPACE controller. For the DC power supply, the Elektro-Automatik PS8360-30 2U (up to 360 $V_{DC}$, 30 A was used). A top-view photograph of the experimental setup is presented in Figure 18.

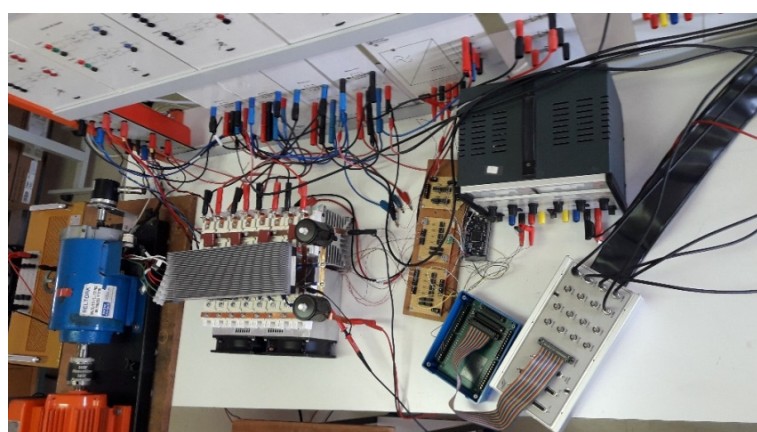

**Figure 18.** Top view photograph of the experimental setup.

Similar tests as the ones implemented for the simulation results were also performed. So, for the first test, an open-switch fault that will affect SRM phase *B* was considered. Thus, the power semiconductor under fault was $S_{12}$. The results that were obtained from this test can be seen in Figure 19. Through these results, it is possible to see the operation of the converter in the healthy and faulty modes. As expected, in faulty mode, the maximum positive voltage level disappears. In this way, the current in the faulty phase does not

reach the reference value. It is also possible to verify that after 30 ms from the beginning of the fault, the converter starts to operate in fault-tolerant mode, recovering its healthy mode. A second test, in which an open-switch fault of a different power semiconductor and a different phase were used, was also performed. In this test, the power semiconductor under fault was $S_{21}$. The obtained results for this test are presented in Figure 20. The results show the three operation modes, healthy, open-circuit failure, and fault-tolerant mode. As expected, when the converter operates in fault mode, the operation of phase A motor winding is severely affected since no voltage is applied. Thus, the current in that winding will always be zero. However, after applying the reconfiguration of the circuit, the motor starts to operate in normal mode, with all voltages and currents having the expected values. The voltage balance between the *DC* capacitors for both open-circuit device failures ($S_{12}$ and $S_{21}$) can be seen in Figure 21. From these figures, it is possible to verify that the balance of the voltage capacitors is achieved in these faulty conditions.

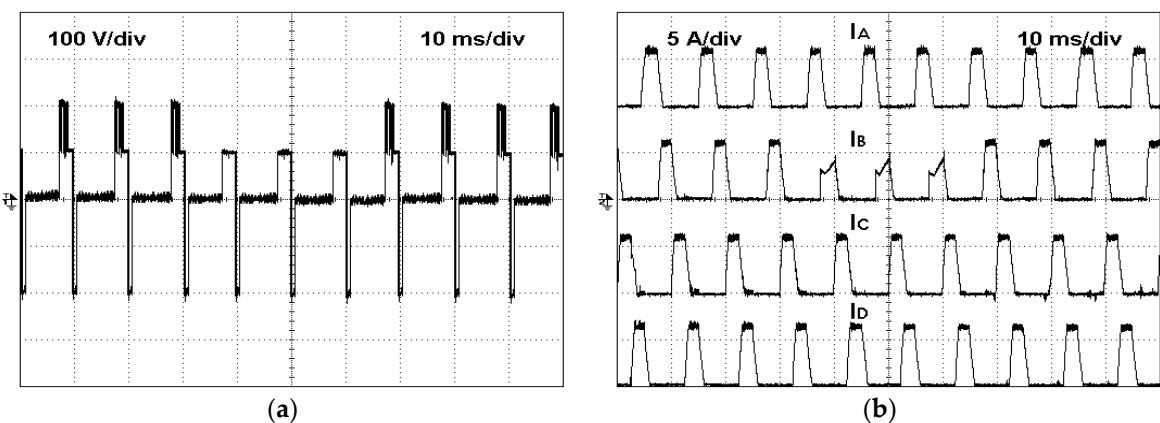

**Figure 19.** Experimental test with the converter operating in normal mode, then with an open-switch fault in the semiconductor $S_{12}$ (**a**) winding voltage of phase *B* and then in fault-tolerant mode; (**b**) winding currents.

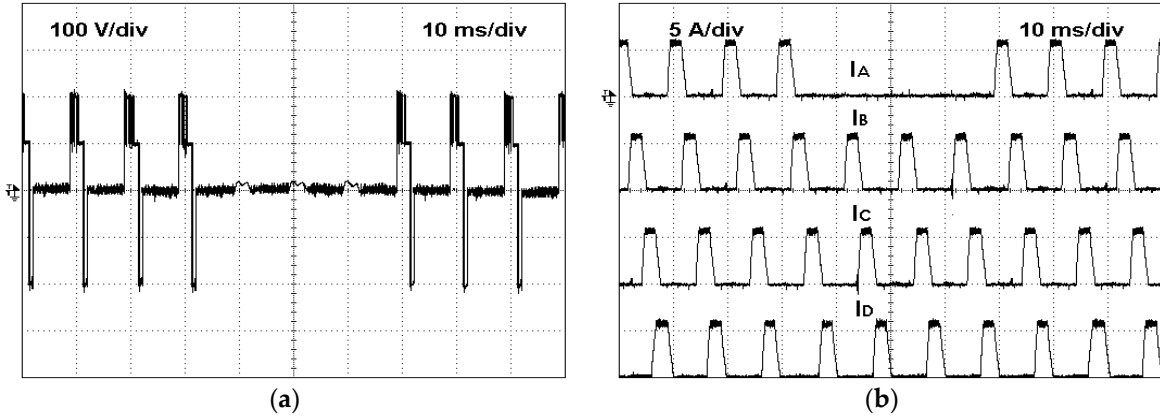

**Figure 20.** Experimental test with the converter operating in normal mode, then with an open-switch fault in the semiconductor $S_{21}$ (**a**) winding voltage of phase *A* and then in fault-tolerant mode (**b**) winding currents.

A different fault type was also tested, namely a short-circuit fault of a power semiconductor. In this test, the power semiconductor under fault was the $S_{24}$. The obtained results of this test are presented in Figure 22. As in the previous case, these results show the three modes of operation, namely healthy, short-circuit fault, and fault-tolerant mode. Thus, when the fault appears, the demagnetization process of the motor winding associated with the leg with the faulty switch is affected. It is not possible to apply the maximum negative voltage by which, during this process, the winding current decreases more slowly. When

the converter starts to operate in fault-tolerant mode (after 30 ms), this problem disappears, as expected, recovering the fast demagnetization. The balance of the *DC* capacitors voltages was also verified during the short circuit of $S_{24}$. This can be seen in Figure 23, where the voltage balance is maintained even after the short-circuit failure.

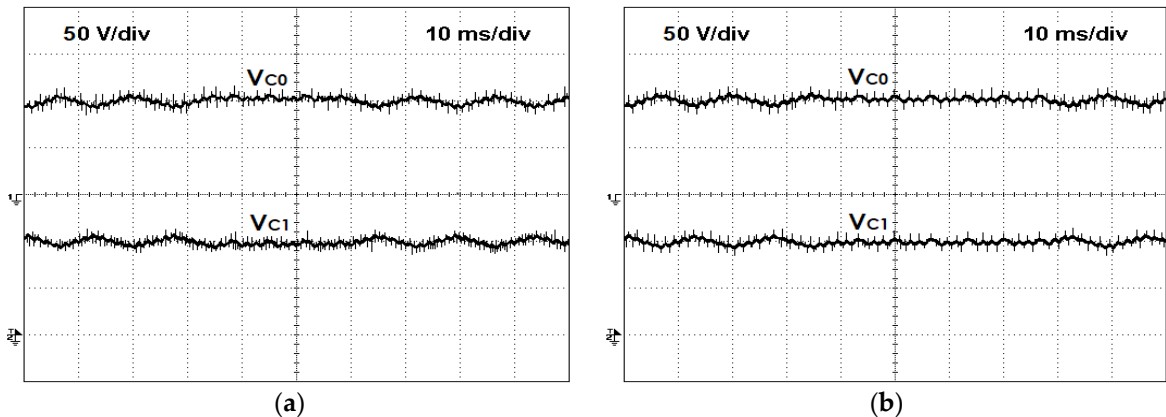

**Figure 21.** Experimental results of the capacitors voltages when the converter is operating first in normal and then with an open-circuit device fault (**a**) open-switch fault in the semiconductor $S_{12}$; (**b**) for open-circuit device fault in the semiconductor $S_{21}$.

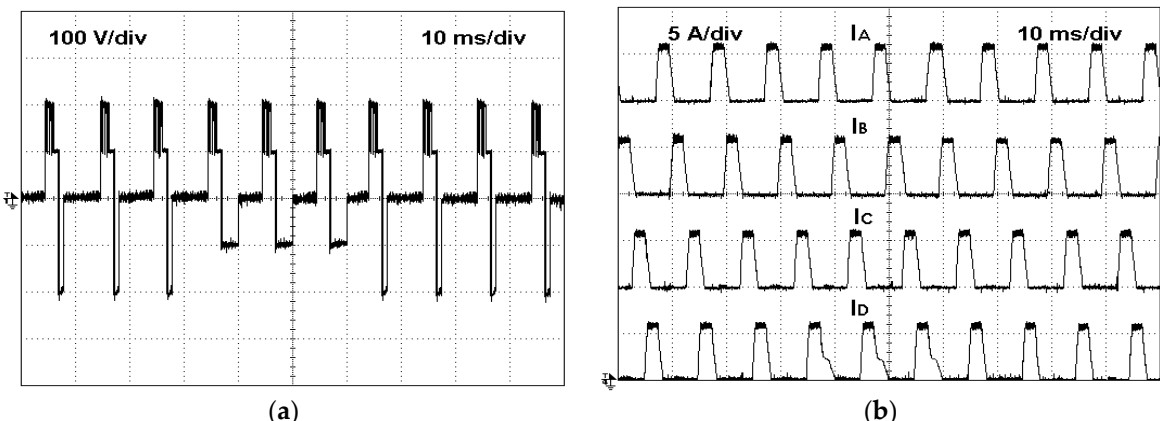

**Figure 22.** Experimental test with the converter operating in normal mode, then with a short-circuit fault in the semiconductor $S_{24}$ and then in fault-tolerant mode (**a**) winding voltage of phase *D*; (**b**) winding currents.

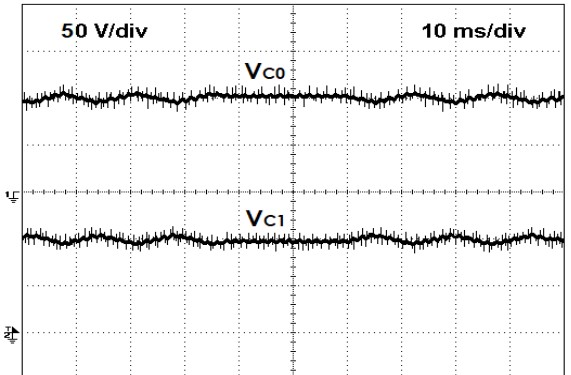

**Figure 23.** Experimental result of the voltages of both capacitors operating in normal conditions and after a short-circuit fault in the semiconductor $S_{24}$.

Another aspect that was verified was the capability to operate in the fault-tolerant mode for multiple faults. Thus, an experimental test in which two faults were considered was also performed. In this case, first, an open-switch fault in semiconductor $S_{11}$ appears, and after some instants, a second open-switch fault in semiconductor $S_{21}$ also appears. The obtained results of this experimental test are presented in Figure 24. Analyzing this figure, it is possible to confirm the two faults. The first fault affects the application of the maximum voltage to the motor winding associated with the faulty leg, by which the current does not reach the desired value. After the second fault, it is not possible anymore to apply any positive voltage to the motor winding associated with the faulty leg. In this way, the current in that winding will always be zero. However, after the reconfiguration process, all those problems disappear, by which all voltage levels are again applied to the affected motor winding. Finally, all the motor currents will follow the references.

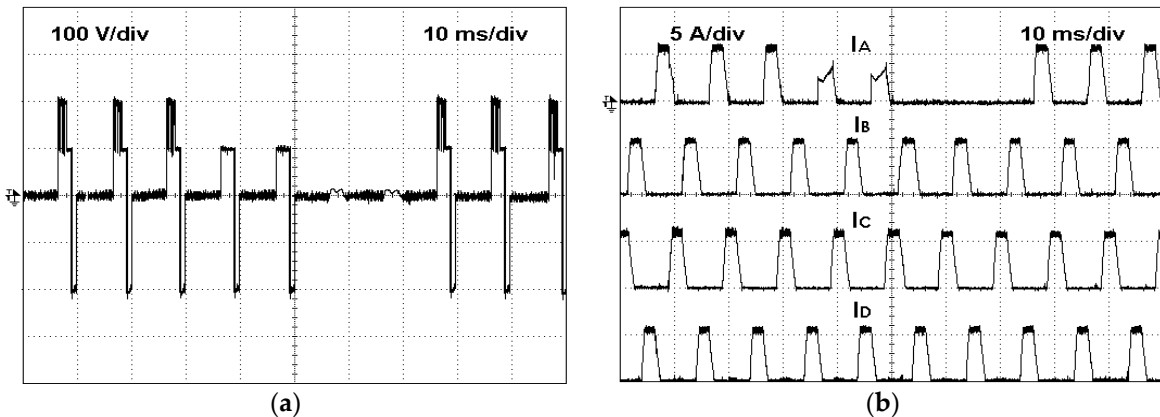

(**a**)  (**b**)

**Figure 24.** Experimental test with the converter operating in normal mode, then with an open-switch fault ($S_{11}$), after that another open-switch fault ($S_{21}$) a finally in fault-tolerant mode (**a**) winding voltage of phase *A*; (**b**) winding currents.

## 7. Discussion

As mentioned in the introduction, several power electronic converters with fault-tolerant capabilities for the SRM were proposed in the literature. The majority of the topologies are characterized by a two-level operation. However, multilevel topologies have also been proposed with the goal to give extended speed operation and reduction of the torque ripple. The disadvantage of these topologies is that usually, they require extra switches and sometimes passive components, such as *DC* capacitors, and this increases the overall cost. Under the point of view of the reliability, this evaluation is not so linear since the two-level topologies with fault-tolerant capabilities require extra switches and sometimes passive components and so do not achieve the same fault-tolerant performance. Let us see an example with a two-level topology with fault-tolerant capability that is considered as one of the references [47]. This topology is characterized by providing fault-tolerant capabilities to all switch faults, namely an open-switch fault and short-circuit fault. Considering, for example, the 8/6 SRM, it requires sixteen switches and four relays. Regarding the multilevel topology presented in [50], it requires sixteen switches but no relays. So, the cost is lower when compared with the previous one. However, it does not provide fault tolerance to all types of faults. The multilevel topology presented in [54] requires sixteen switches. However, in the case of the existence of a switch failure, immediately lost is one of the voltage levels, which could increase the ripple and limit the maximum required speed. Another limitation of this topology is that in the case of a switch short-circuit, there appears an overcurrent. The multilevel topology given by [53] requires thirty-two switches but no relays. It practically solves all the switch faults. However, the number of switches is much higher than the two-level solution. On the other hand, in a particular case, the operation is not completely restored (provides fault tolerance but

without one of the voltage levels). Regarding the proposed multilevel topology, it requires thirty-four switches and eight relays. Thus, from the point of view of components, it requires a higher number of switches and relays. Due to that, the cost and complexity of the proposed solution is higher than the other ones. However, this is the unique solution that provides a complete fault-tolerant operation to any kind of switch fault with multilevel characteristics. On the other hand, this topology also provides fault-tolerant capabilities to a higher number of multiple faults.

## 8. Conclusions

One important aspect associated with the SRM is the power electronic converter. This paper presents a new fault-tolerant power converter for the 8/6 SRM. The proposed fault-tolerant converter was designed in a way that is able to handle two different types of faults, namely, open- and short-circuit ones. Moreover, it is also able to handle multiple faults in some conditions. Apart from its inherent fault-tolerant capability, this power electronic converter is also able to provide multilevel voltages to the SRM being, in this way, indicated to be used in applications that require high-speed range. The operation of the converter in normal and fault conditions was also addressed in a detailed way. The study of the proposed solution was also verified through computer simulations and laboratory tests from a developed prototype. From the acquired results based on these tests, it was possible to verify the capability of the proposed solution to operate in normal and fault conditions without loss of performance.

**Author Contributions:** Conceptualization, V.F.P. and A.J.P.; methodology, V.F.P.; validation, A.C. and D.F.; formal analysis, V.F.P., A.C. and A.J.P.; investigation, V.F.P. and A.C.; writing—original draft preparation, V.F.P. and A.J.P.; writing—review and editing, D.F. and A.J.P.; visualization, D.F. and A.J.P.; supervision, A.J.P. All authors have read and agreed to the published version of the manuscript.

**Funding:** This research was funded by FCT Funda?ão para a Ciência e a Tecnologia grant number UID/CEC/50021/2020 and UID/EEA/00066/2020.

**Data Availability Statement:** Data sharing not applicable.

**Acknowledgments:** This work was supported by national funds through FCT Funda?ão para a Ciência e a Tecnologia with reference UID/CEC/50021/2020 and UID/EEA/00066/2020.

**Conflicts of Interest:** The authors declare no conflict of interest.

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
