# Peer review of "Fault-Tolerant Multilevel Converter to Feed a Switched Reluctance Machine"

_machines, doi:10.3390/machines10010035_

Round 1
Reviewer 1 Report
In this paper the authors present a fault tolerant converter, based on a Neutral Point Clamped Asymmetric Half-Bridge (NPC-AHB) converter applied in a SRM drive.
The authors should enlarge the bibliographic review.
The authors should improve the written English, in particular the final sentence of page 10.
The authors must be coherent in the designation of the switches, in page 12 the author refer to switch S1A, however in Fig. 2 to 6 there are not a switch S1A.
The authors must present the rationale for the definition of the prioritization presented in Table IV.
The authors must present, in detail the fault diagnostic approach and the reason why the time to for the system reconfiguration is so high (several phase periods).
The authors should provide evidence of the ability of the converter to keep the capacitors voltages balanced. In the simulation results the authors feed the motor with two DC voltage sources avoiding dealing with this issue.
The authors should present a photo of the experimental setup.
The experimental phase current waveforms must be easily identified. I recommend presenting each phase current in a different window.
To obtain fault tolerance, the authors add four switches per phase, plus two mechanical relays per phase and two additional switches to isolate the neutral point. Please present a cost/benefit analysis comparing the proposed solution with other fault tolerant converters. It is the reviewer opinion that other solutions can provide similar fault tolerance with less complexity.
Author Response
First of all, we would like to take this opportunity to express our gratitude to the reviewers and the editorial board for their valuable remarks and comments on the manuscript. All remarks and comments have been applied in the manuscript and answers to reviewers are also given hereafter in detail.
Author’s Response To Comments Of The Reviewer#1
Comment n.º 1 of Reviewer1:
In this paper the authors present a fault tolerant converter, based on a Neutral Point Clamped Asymmetric Half-Bridge (NPC-AHB) converter applied in a SRM drive.
The authors should enlarge the bibliographic review.
Response to Comment n.º 1 of Reviewer1:
Thank you very much for your analysis of our work and respective comments. You are right, we should enlarge the bibliographic review. So, in accordance, we enlarged and improved the introduction section, namely the state-of-the-art, including more references.
Comment n.º 2 of Reviewer1:
The authors should improve the written English, in particular the final sentence of page 10.
Response to Comment n.º 2 of Reviewer1:
You are completely right. A new revision was made to improve the written English.
Comment n.º 3 of Reviewer1:
The authors must be coherent in the designation of the switches, in page 12 the author refer to switch S1A, however in Fig. 2 to 6 there are not a switch S1A.
Response to Comment n.º 3 of Reviewer1:
Thank you very much for noticing this lapse. You are completely right. This problem was now solved in this revision, changing S1A to S11.
Comment n.º 4 of Reviewer1:
The authors must present the rationale for the definition of the prioritization presented in Table IV.
Response to Comment n.º 4 of Reviewer1:
You are completely right. The prioritization presented in Table IV was not clear, indeed. Additional justifications and details were introduced in the manuscript (section 3) to clarify this aspect.
Comment n.º 5 of Reviewer1:
The authors must present, in detail the fault diagnostic approach and the reason why the time to for the system reconfiguration is so high (several phase periods).
Response to Comment n.º 5 of Reviewer1:
Thank you very much for pointing out this issue. In the manuscript, we didn’t introduce the fault diagnostic approach since the purpose was only to introduce the new fault‑tolerant topology. The reason for the several phase periods was intentional to show clearly the three possible conditions, namely the healthy, fault operation and fault tolerant operation. For this topology, any fault detection methods that were also proposed, can be used. However, we agree with the reviewer that with the introduction of a fault detection algorithm the paper will be improved. In this way, the fault tolerant detection algorithm developed by the authors for multilevel converters applied to SRM was introduced in section 4. We decided to maintain in simulations and experimental results the time interval between the three conditions to clearly show the change from healthy, to fault and finally to fault-tolerant operation. Otherwise, it would be difficult to clearly see the behaviour of the converter during these steps.
Comment n.º 6 of Reviewer1:
The authors should provide evidence of the ability of the converter to keep the capacitors voltages balanced. In the simulation results the authors feed the motor with two DC voltage sources avoiding dealing with this issue.
Response to Comment n.º 6 of Reviewer1:
The reviewer is completely right. So, we presented new results to demonstrate the voltage balance of capacitors. The obtained results show that the capacitor voltages can remain balanced in both normal and faut-tolerant operation. Moreover, we introduced additional details of the voltage balance process in section 3 to clarify all the related aspects. In this way, we think that this issue becomes clearer now, showing that this balance can be achieved even in fault-tolerant operation. Several simulations and experimental were also introduced to prove this ability and provide evidence about this issue.
Comment n.º 7 of Reviewer1:
The authors should present a photo of the experimental setup.
Response to Comment n.º 7 of Reviewer1:
Thank you very much for this suggestion. A photograph of the experimental setup is now available at the beginning of section 6.
Comment n.º 8 of Reviewer1:
The experimental phase current waveforms must be easily identified. I recommend presenting each phase current in a different window.
Response to Comment n.º 8 of Reviewer1:
Thank you very much for this suggestion. In this revision, we decided to separate the current waveforms of the experimental results to easily identify the differences of each phase. In the simulation results this is easy to undestand since they have different colours.
Comment n.º 9 of Reviewer1:
To obtain fault tolerance, the authors add four switches per phase, plus two mechanical relays per phase and two additional switches to isolate the neutral point. Please present a cost/benefit analysis comparing the proposed solution with other fault tolerant converters. It is the reviewer opinion that other solutions can provide similar fault tolerance with less complexity.
Response to Comment n.º 9 of Reviewer1:
This is very well highlighted. We completely agree with the reviewer in which this solution requires high number of switches and relays. However, the proposed topology is one (under the point of view of the topologies) that provides multilevel operation with the best fault tolerant capability. Most of the multilevel topologies found in the scientific literature only provide some fault-tolerant capability since they cannot handle with some fault types, such as short-circuits in power devices. Even the best multilevel topology under the point of view of the reliability [47] presents a limitation since the full operation is not completely restored (provides fault tolerance but without one of the voltage levels). Regarding the proposed multilevel topology, this is the unique solution, known by the authors, that provides a complete fault-tolerant operation to any kind of switch fault with multilevel characteristics. On the other hand, this topology also provides fault tolerant capability to multiple faults. So, as any topology, this one has advantages and disadvantages, being necessary to study which criteria is the most important. Thus, we think that the presentation of this new topology is important to give more options to the designers. However, we have the same opinion of the reviewer that this discussion is very important and should be in the paper. In this way, we introduced a new section 7 in which we discuss this issue.
Reviewer 2 Report
1. What measuring instruments were used for research?
2. How the results obtained can be generalized to electric drives with power P>500 kWt?
Author Response
First of all, we would like to take this opportunity to express our gratitude to the reviewers and the editorial board for their valuable remarks and comments on the manuscript. All remarks and comments have been applied in the manuscript and answers to reviewers are also given hereafter in detail.
Author’s Response To Comments Of The Reviewer#2
Comment n.º 1 of Reviewer2:
What measuring instruments were used for research?
Response to Comment n.º 1 of Reviewer2:
Thank you very much for your analysis and comments. Several instruments and sensors were used in this research. Regarding the instruments, an oscilloscope TDS3014C with 100mV/A current probes and voltage probes with 1V/100V. Regarding the sensors, four 20A (LEM HSX20-NP) current sensors were used to measure the current of each phase and send the value for the DSPACE1104 which controls the speed of the motor. The position of the SRM rotor is measuring by using an absolute encoder (TWK DAB 66 - M 360 W C01 with 0-10V per revolution) which signal was also sent to the DSPACE controller. For the DC power supply, it was used the Elektro-Automatik PS8360-30 2U (up to 360VDC, 30A). All this information was introduced in section 6 of the revised paper.
Comment n.º 2 of Reviewer2:
2 - How the results obtained can be generalized to electric drives with power P>500 kW?
Response to Comment n.º 2 of Reviewer2:
Thank you very much for this comment. In theory, it is always possible to adapt the proposed solution for high power electric drives such as the proposed by the reviewer. Nevertheless, to achieve P>500kW it will be necessary to take care about the desired operational voltage and currents. For drives operating with high voltages, the proposed topology should be modified to a higher number of voltage levels since there are limitations about the maximum reverse voltage applied to power devices (currently about 1700V maximum for most IGBTs technology). Increasing the number of IGBTs connected in series in multilevel structures it is possible to operate over 10kV. The same principle can be adopted to high currents in electric drives (e.g. 350A), introducing multiple parallel branches or power devices to withstand the maximum desired current. These principles have been widely used over the last years by several known manufactures to produce high power electric drives. Obviously, the use of higher voltage and currents introduce additional problems regarding EMC (electromagnetic compatibility) emissions and isolation problems that should be treated with care. Several improvements and reinforcements in the design stage should be considered to avoid additional problems in the drive circuits, PCBs, connectors, heat sinking and other elements. Some comments related with this question were introduced in section 2 of the paper.
Round 2
Reviewer 1 Report
In this paper the authors present a fault tolerant converter, based on a Neutral Point Clamped Asymmetric Half-Bridge (NPC-AHB) converter applied in a SRM drive.
The authors should check the citations, in the text, of the bibliographic references. Some citations don’t match. Considering that the motor of this work is a 4-phase motor, for the Park’s Vector approach the author should cite works where the Park’s Vector is applied in the fault diagnostic of 4-phase SRM.
Apart from that, the authors addressed all of the significant issues raised by the reviewer. Therefore, I believe the paper should be accepted.
Author Response
First of all, we would like to express our gratitude to the reviewer for their valuable remarks and comments on the manuscript. We also like to thank for the positive feedback to our changes
Reviewer Comment:
The authors should check the citations, in the text, of the bibliographic references. Some citations don’t match. Considering that the motor of this work is a 4-phase motor, for the Park’s Vector approach the author should cite works where the Park’s Vector is applied in the fault diagnostic of 4-phase SRM.
Response to comment:
You are completely right. Indeed some citations do not match. In this way we change the number of the some citations on the text, as well as, the order that appears in the references in order to match. The references that were affected was from 60 to 69. We also verify that the question about the works of Park vector related with diagnosis in SRM. We check again works about that in the scientific databases, and we found a new work in which parks vector is applied in the fault diagnostic of 4-phase SRM. In this way, we introduce a new reference (64) in order to improve the introduction with important references. We expect that we have answered the questions that still remained.